# Lentiviral Vectors for Ocular Gene Therapy

**DOI:** 10.3390/pharmaceutics14081605

**Published:** 2022-07-31

**Authors:** Yvan Arsenijevic, Adeline Berger, Florian Udry, Corinne Kostic

**Affiliations:** 1Unit Retinal Degeneration and Regeneration, Department of Ophthalmology, University of Lausanne, Jules-Gonin Eye Hospital, Fondation Asile des Aveugles, 1004 Lausanne, Switzerland; florian.udry@fa2.ch; 2Group Epigenetics of ocular diseases, Department of Ophthalmology, University of Lausanne, Jules-Gonin Eye Hospital, Fondation Asile des Aveugles, 1004 Lausanne, Switzerland; adeline.berger@fa2.ch; 3Group for Retinal Disorder Research, Department of Ophthalmology, University of Lausanne, Jules-Gonin Eye Hospital, Fondation Asile des Aveugles, 1004 Lausanne, Switzerland

**Keywords:** lentivirus, viral vector, gene therapy, ocular delivery, retinal disease, glaucoma, cornea

## Abstract

This review offers the basics of lentiviral vector technologies, their advantages and pitfalls, and an overview of their use in the field of ophthalmology. First, the description of the global challenges encountered to develop safe and efficient lentiviral recombinant vectors for clinical application is provided. The risks and the measures taken to minimize secondary effects as well as new strategies using these vectors are also discussed. This review then focuses on lentiviral vectors specifically designed for ocular therapy and goes over preclinical and clinical studies describing their safety and efficacy. A therapeutic approach using lentiviral vector-mediated gene therapy is currently being developed for many ocular diseases, e.g., aged-related macular degeneration, retinopathy of prematurity, inherited retinal dystrophies (Leber congenital amaurosis type 2, Stargardt disease, Usher syndrome), glaucoma, and corneal fibrosis or engraftment rejection. In summary, this review shows how lentiviral vectors offer an interesting alternative for gene therapy in all ocular compartments.

## 1. Basics of Lentiviral Vector Technology

### 1.1. From Lentivirus to Lentiviral Vector

Lentiviruses belong to the retroviradae family characterized by an RNA genome (ssRNA of 8 to 10 kb) packed with an integrase, a reverse transcriptase, and a protease within a capsid (p24 protein) itself included in an envelope made of lipids of the host membrane (acquired during the budding process) and viral glycoproteins. Compared to other retroviruses, lentiviruses have the ability to infect non-dividing cells thanks to the import of their pre-integration complex to the nuclear compartment [1]. This property rapidly promoted interest in the development of recombinant lentiviral vectors for gene transfer. 

Five lentiviral serogroups exist, depending on the targeted vertebrate host. Human immunodeficient virus 1 (HIV-1), equine infectious anemia virus (EIAV), simian immunodeficiency virus (SIV), feline immunodeficiency virus (FIV), or bovine immunodeficiency virus (BIV) were thus engineered with the same strategy used to develop recombinant vectors from native viruses. First, a large portion of the viral genome containing the genes encoding for the viral structure and replication processes was deleted. Several rounds of engineering improvements, already precisely described in Dropulic et al. [2] and discussed in Section 1.2.2, increased the safety of these vectors allowing stable expression in dividing cells. The third generation of lentiviral vector (LV) necessitates four plasmids, among which the transfer plasmid carries the therapeutic cassette. This plasmid contains an up to 10 kb sequence template of the recombinant genome consisting of conserved key viral sequences such as 5′LTR (long terminal repeat), 3′LTR modified as a self-inactivating LTR (SIN), the major splice site donor and acceptor, the packaging signal sequence (Ψ) and Rev response element (RRE), the nuclear import signal (central polypurine tract, cPPT) [3], and the transgene cassette (foreign sequences to be transferred). The role of each of these elements will be more precisely described below (Section 1.2 and Section 1.3). 

Second, the essential viral proteins required for particle assembly were also identified in order to express them in *trans* using a packaging plasmid or stable cell lines for the production step. *Trans*-complementation during the production thus generates infectious but not pathogenic nor replicative recombinant particles (Section 1.3). 

Third, a major modification of lentivirus was the modification of its targeting. Initial works used the native viral envelop encoded by the viral env gene, which, for HIV-1, is selective to CD4+ cells [4,5]. However, the discovery that recombinant lentiviral vectors, like other retroviruses, can be pseudotyped with the vesicular stomatitis virus G (VSV-G) glycoprotein, which has a broad tropism opened the use of this vector to a wide field of applications [1,6,7,8] and further targeting developments (Section 1.4).

### 1.2. Lentivirus and Lentiviral Vector Integration 

On top of providing a stable expression due to genome integration, a low immune response, a broad tropism, and a large cloning capacity, lentiviral vectors also have, as mentioned above, the undeniable advantage over other retroviral viruses to be able to transduce non-dividing cells. Nevertheless, if some lentivirus particularities can be seen as a clear benefit to broaden the range of gene therapy applications, they become a threat in a safety context [9]. Indeed, since its first considerations as a clinical tool, lentiviral vectors have raised three major concerns: (1) their replication ability with the risk of producing replication-competent lentiviruses within the host cells, (2) the risk of recombination at the post-transcriptional level, and (3) the insertional mutagenesis potential associated with its integration ability. 

We will briefly describe here the different components of a lentiviral vector, their properties, the risks associated with LV use in human therapeutic approaches, and past and recent strategies developed to improve the safety of these vectors and reach a safety level compatible with human clinical applications.

#### 1.2.1. Components and Functioning of Lentiviral Infection

To efficiently transduce a target cell, the lentiviral genome is composed of the following coding genes [10]:Gag encoding for matrix (MA), nucleocapsid (NC), capsid (CA), and p6 proteins required for viral assembly and infection;Pol encoding for reverse transcriptase (RT), RNase H, protease (PRO), and integrase (IN), essential for reverse transcription and genomic integration;Env encoding for surface glycoproteins (SU, TM) that define tropism of the virus and enable entry into the host cells;Tat and Rev, two regulatory genes that activate viral transcription and nuclear export of intron-containing nascent viral RNA, respectively;Four accessory genes, Vif, Vpr, Vpu, and Nef;Several *cis*-acting elements are also critical for viral replication, such asψ, the packaging signal;RRE, the Rev response element, required for processing and transport of viral RNAs;cPPT/CTS, the central polypurine tract, and the central termination sequence, whose role is still questioned, suspected to be linked to nuclear transport and replication;TAR, the TAT activation region;The splicing donor (SD) and acceptor (SA) sites, which allow the production of all viral proteins and the viral RNA starting from a unique pre-mRNA.

The insertion of the viral genome into the host genome follows the subsequent steps: (1) binding to the host cell, membrane fusion, and entry of the capsid, (2) reverse transcription of the viral RNA into DNA, (3) nuclear import, and (4) provirus integration as depicted in Figure 1. The uncoating of the capsid is still not clear as CA was shown to influence reverse transcription, nuclear import, and provirus integration [11].

Integration is mediated by IN encoded in the viral pol gene and cleaved by PRO from the Gag-Pol polyprotein [12]. The integration process is the result of several steps: vector 3′-end modification, opening of host genome and insertion of viral DNA, gap repair, and ligation [13]. Genomic integration is a relatively inefficient process, which is unequal regarding the location of integration within the host genome. It has been previously shown that lentiviruses, unlike other retroviruses, preferentially integrate into active transcription units, the integration site being supposedly influenced by chromatin accessibility, cell cycle effects, and tethering mechanism [14]. Indeed, it was shown that lens epithelium-derived growth factor (LEDGF/p75), a host cell protein, is able to target the pre-integration complex to DNA and promotes integration [15,16,17]. Once integrated, the 5′LTR serves as a promoter to drive the viral genome transcription, using the host transcription machinery. First, several viral proteins will be synthetized and, in association with *cis*-acting elements, finely regulate both transcription and splicing efficiencies, to drive the successive production of all viral proteins and unspliced viral mRNAs from the same viral genome. 

The viral processes, from the entry into the host cells to the import of the viral genome into the nucleus, are some of the important features needed to adapt this virus as a gene therapy tool (Figure 1). If these endogenous lentivirus abilities clearly serve as advantages for clinical transfer to gene therapy, its replication capacity must be inhibited to ensure the safe use of LV. Indeed, since lentiviruses divert the host machinery for their life cycle and replication, it rapidly induces the apoptosis of the host cell and leads to an aberrant release of new lentiviral particles. While the strategy of producing replication-incompetent LV, described in Section 1.2.2, has been rapidly adopted in the field of gene therapy, the risk of recombination with wild-type lentivirus still exists [18]. Moreover, recombination at the post-transcriptional level, for example, between splicing sites of the viral RNA and the host RNAs, is another treatment to consider [19], even if it can sometimes be beneficial for the treatment [20]. 

The property of retroviruses to integrate into the host genome is undoubtedly an advantage for retroviral vectors to drive a stable and long-term expression of the integrated cassette. Nevertheless, this comes at a cost. Depending on the exact location of integration, the viral sequences can impact the surrounding host genes. Four different mechanisms have recently been described as retroviral insertional mutagenesis in humans [21]. While gene transactivation following viral genome insertion has rarely benefited the cells, providing, for example, a proliferative advantage to the transduced cells [22], it is more often seen as a threat. Indeed, since one of its first use in human clinics, gene therapy using retroviruses (Moloney retrovirus), despite showing at first an impressive efficiency, is unfortunately followed by a lethal risk of mutational insertion [23,24]. Several challenges have arisen, such as the determination, the prediction, or even the control of the integration of the viral sequence to avoid transactivation of an oncogene or inhibition of a tumor suppressor.

Several approaches to determine both the integration site and the vector copy number per cell have been developed and recently improved, ranging from optimized linear amplification-mediated PCR (LAM-PCR) [25] to enhanced-specificity tagmentation-assisted PCR (esTag-PCR) [26] or digital droplet PCR (ddPCR), associated with the use of lentiviral cellular standards [27].

Moreover, the oncogenic potential of LV has also been evaluated and compared to that of recombinant adeno-associated virus (rAAV) in vivo, after subretinal injection in p53^−/−^ and p53^+/−^ (longer lifespan than p53^−/−^) mice, highly susceptible to intraocular malignant transformation. High concentrations of rAAV.CMV.hrGFP (1 × 10^12^ vg mL^−1^), rAAV.hRPE65p.hRPE65 (1 × 10^11^ vg mL^−1^) and VSV-G pseudotyped HIV-1.SFFV.hrGFP vectors (5 × 10^8^ TU mL^−1^) did not promote ocular tumor growth in any of the animals tested [28]. This indicates that, unlike other retroviruses (gammaretroviral vectors [29]), LV insertional mutation event is a risk low enough to support the safety of LV for clinical use. This is even reinforced in the context of ophthalmic applications by the post-mitotic status of the targeted cells, which reduces the risk of accumulation of mutational events.

More recently, impacted by the success stories of the CRISPR-Cas9 system in gene editing, LV-based ocular gene therapy has seen the arrival of an LV carrying both a Cas9 mRNA and a Vegfa-targeting guide RNA to treat wet age-related macular degeneration (AMD) [30]. The study conducted in mice included another level of safety assessment, which is the validation of no Cas9 off-target events when analyzing the in silico-most likely identified sites using deep sequencing. 

#### 1.2.2. Strategy to Limit the Risks and Increase LV Safety

Two main approaches are possible to reduce at maximum the safety risks of using LV for gene therapy: first, to remove any non-essential viral sequences for transduction and expression from the construct, and second, to limit viral genome integration into the host cellular genome. 

Regarding the first approach, three generations of lentiviral packaging systems have been designed with the goal of reaching a safety level acceptable for therapeutic application. Only the second and third generations are still currently used with a clinical perspective. The second generation proposed to split the lentiviral vector components across three plasmids, the transfer plasmid, the packaging plasmid, and the envelope plasmid. The third generation goes further by splitting the packaging plasmid into two, increasing the complete system to four plasmids (Figure 2).

The addition of an internal promoter (CMV is the most used one) eliminates the need for the tat gene. Additionally, this third generation also includes an important modification on the 3′LTR, the deletion of its U3 region, consisting of enhancer and promoter sequences. The deletion of the promoter/enhancer regions of the LTRs prevents the *cis*-activation of genes adjacent to the integration site of the viral DNA. Moreover, during the reverse transcription step, the deletion is transferred to the 5′LTR of the proviral DNA, rendering the viral vector self-inactivating (SIN, and thus unable to replicate) [31,32,33,34]. The final third-generation system (Figure 2), driving LV enable to replicate but still capable of integration, is thus composed as follows:The transfer plasmid that contains the transgene and its promoter, the 5′ and ΔU3–3′ LTRs, the ψ sequence, and the RRE sequence;One packaging plasmid encoding gag and pol;A second packaging plasmid encoding rev;The envelope plasmid encoding the envelope proteins, which provide the vector tropism (see Section 1.4).

Optionally, to increase the transgene expression efficiency, the woodchuck hepatitis virus post-transcriptional regulatory element is often introduced in 3′ of the transgene [35]. The transgene plasmid can also be used to transfer two genes either separated by the IRES element, which promotes the translation of a second protein from a unique mRNA [36], or by the 2A element, which mediates the cleavage of two proteins from the same mRNA [37]. 

The reduction in lentiviral genes from nine to three in the packaging system was a first step to limiting the consequences of viral transduction. The idea of reducing the lentiviral components to only essential sequences was also applied to EIAV lentiviral vector system (pONY 8.4 and 8.7 vectors) [38]. The pONY8.4 is self-inactivating (SIN), rev-independent, and composed only of EIAV essential *cis* elements. The pONY8.7 genome also incorporates the cPPT and woodchuck hepatitis virus post-transcriptional regulatory element (WPRE) elements to increase the efficiency of transduction and expression levels.

During LV production, homologous recombination of overlapping sequences between the different plasmids could generate recombinant-competent lentiviral vectors (RCLs), which are replicative vectors. Many studies aimed to develop sensitive and reliable tests to quantify this risk [39,40,41,42,43]. In addition, other systems were developed to minimize RCLs, such as Rev-independent LV [44] or a five plasmids *trans*-lentiviral system, which separates the RT-IN from the Gag-Pol [45,46]. However, even if the approach of isolating the RT-IN in a supplementary plasmid is interesting, the presence of some accessory genes in the packaging system is questionable for clinical application. Moreover, no RCLs were identified with the third generation of LV, validating the use of the four plasmid systems for therapeutic purposes [42,47].

To avoid at maximum the events of recombination with RNAs of the host cell, Vink et al. proposed to move the ψ and RRE sequences downstream of the SIN 3′LTR in the LTR1 LV [48]. In this way, these sequences still participate in the processing of the viral particles but are not reverse-transcribed and thus absent from the delivered provirus. 

The second approach consists in depriving the LV of its integration ability, which prevents integrated viral DNA and increases instead the level of circular episomes in the host cell. The system used for LV production will then be based on third-generation *IN*-deficient packaging plasmids. If this can be seen as safer regarding the insertional mutagenesis risk, it also subtracts from the system its property to maintain transgene expression in dividing cells. This transient expression can be seen as an advantage in some conditions, such as CRISPR/Cas9-mediated gene editing [49], vaccination, or induction of recombination [50]. Additionally, the use of integrative-deficient LV (IDLV) can also be an efficient alternative for non-dividing cells [51,52]. Two main strategies are possible to generate IDLV: (1) by introducing a mutation into the pol gene to specifically mutate the integrase without disturbing the other properties of IN or the other proteins arising from the same gene, or (2) by mutating the integrase DNA attachment site [53,54]. Nevertheless, some integrase-independent integration events have been reported using IDLV [55,56].

### 1.3. Lentiviral Vector Production

#### 1.3.1. Lentiviral Vector Assembly

To produce recombinant lentiviral particles, expression in *trans* of essential structural and enzymatic viral proteins is needed. Synthesis of these proteins will allow the assembly of viral particles (Figure 2). Some sequences of the viral genome are required to pack the transgene cassette into the recombinant viral particle and must then be conserved in the transgene plasmid. First, the recombinant genome is delimited by the LTR, which was modified to increase the safety of the lentiviral vector (as described above Section 1.2.2). Replacement of the HIV-1 promoter sequence of the 5′LTR by a CMV promoter also increases the amount of transcription of the recombinant genome and thus the recombinant viral particle production [32,33,57]. Then the SA, SD, cPPT, and RRE sequences are essential for the preparation and packaging of the transfer cassette into the recombinant viral particle and its import into the host cell nuclei [3,7].

Following the third generation of lentiviral packaging systems, the complementation for recombinant viral vector production is performed by co-transfection of the transfer plasmid containing the recombinant genome with two packaging plasmids encoding the different proteins necessary to generate viral particles and a plasmid encoding the appropriate envelope. These transfections are usually performed in HEK-293T cells that are highly transfectable and express the SV40 large-T antigen, a protein allowing replication of plasmids bearing SV40 origin. Even if productions in these cells are very efficient, alternative cell lines were explored especially to scale the production up [58,59,60]. Transfection of packaging plasmid(s) is widely used in laboratories for basic or preclinical research but also to prepare vectors for clinical applications [61]. Other significant works were made to reach large-scale production using the transfection methods, see Segura et al. [62]. The use of baculovirus to transfer the different constructs is also a successful alternative for LV production [63,64]. 

Cell lines stably expressing the packaging system were generated to avoid the important consumption of packaging plasmids, to prevent potential contamination of the vector preparation with those plasmids, and to reduce the risk of recombination within these plasmids and the transfection variation. Since the expression of the packaging plasmids is toxic for the cells, the expression of these proteins under the control of an inducible system allows producing vector particles over many days (reviewed in Martinez-Molina et al. [65]). Even if the establishment of this type of stable cell line requires a certain investment, it appears to be a pertinent approach for the large-scale manufacturing of approved vectors [66].

#### 1.3.2. Recombinant Lentiviral Preparation and Purification

The initial common method of LV batch preparation in a research laboratory is based on the particularity of the VSV-G envelope to resist ultracentrifugation forces. Therefore, crude extracts can be prepared after filtration of the cell culture supernatant and digestion of plasmid DNAs by two successive centrifugations at 65,000× *g*. A high concentration of lentiviral particles (>10^8^ TU/mL), especially needed for in vivo experiments, can be reached with this method. However, this kind of preparation still contains contaminations from the cell culture, which can induce some in vivo toxicity. Moreover, since certain envelopes cannot tolerate ultracentrifugation, alternative vector purification protocols compatible with large-scale production and improving the purity of the preparation have been implemented. First, the clarification to eliminate cell debris is based on successive microfiltrations from 0.45 to 0.2 µm pore to avoid clogging and loss of particles. Then, concentration and purification are performed using tangential flow filtration (TFF) and/or chromatography. TFF is based on size exclusion using membranes with 1 to 100 nm pores to separate serum proteins, DNA fragments, and other cell culture contaminants [67]. On the other hand, chromatography relies on the envelope properties to purify recombinant particles. Anion-exchange chromatography uses a cationic column to select negatively charged LV particles; however, a high salt concentration is then needed to recover the particles, and this treatment can affect the transduction efficiency of the vector [68]. Affinity chromatography, on the other hand, specifically catches the particles using a ligand-receptor interaction with one of the envelope proteins. LVs are mainly collected using a heparin column, but the elution is also dependent on salt concentration and requires an additional step such as a TFF for buffer exchange [58,69]. For large-scale preparation, a final size-exclusion chromatography allows to discard the contamination with small elements and isolate the larger vector particles [70].

LV production and purification have been optimized to answer to the constraints of scale-up and good manufacturing process (GMP) conditions to produce LV clinical batches: from high capacity cell culture systems for adherent cells (i.e., cell factory) or suspension cells (bioreactor) to adapted downstream process (i.e., chromatography, filtration, for a detailed review on the latest technologies Shi et al. 2022 [71]).

#### 1.3.3. Clinical Preparation

More precisely, in the context of ocular diseases, clinical preparations of vectors were mostly conducted by co-transfection of three or four plasmids on HEK-293T cells. Upscaling was reported with the use of a cell factory [72,73] or Bioreactor [74], sometimes including several transfection adaptations such as the addition of polyethylenimine (PEI) and sodium butyrate [74], use of lipofection [75]. RetinoStat, Stargene, and UshStat (LV discussed more in detail in Section 2) were produced in GMP conditions using anion-exchange membrane chromatography and hollow fiber tangential flow ultrafiltration/diafiltration [36,76,77]. Each corresponding titer was defined by measuring the reverse transcriptase activity (product-enhanced reverse transcriptase (PERT)) and RNA copy number using qRT-PCR [36,78]. The safety profile of these vectors was demonstrated in rabbit and non-human primate (NHP), which led to human application in clinical trials (see chapter II). Similar preparation was used for a GMP-like HIV-derived vector injected in NHP [74]. However, LV productions performed by ultracentrifugation were also well tolerated in several other in vivo conditions. SIV-hPEDF was evaluated in NHP following production based on co-transfection of four plasmids and ultracentrifugation-based concentration [79,80]. The toxicological profile was evaluated as promising with minimal effect on retinal function (measured by electroretinogram (ERG)) and inflammation while presenting a long-term expression [75,81]. The ultracentrifugation method also provided suitable tolerance after LV [82,83] or FIV [72,84] injection in the anterior chamber of NHP. Following successful use for ex vivo approaches, mainly for hematopoietic diseases [85], the continuous development of lentiviral production and safety assessment now provide reliable tools to make LV attractive for in vivo clinical application.

### 1.4. Envelope

Optimization of the vector targeting is a crucial step to improve its efficiency and thus reduce the therapeutic dose and concomitant adverse effects. Researchers explored the possibility of using alternative envelopes to generate recombinant vectors in a process called pseudotyping. These pseudotyped vectors then benefit from the properties of their foreign envelope, which can influence the cell-specific targeting, the entry mechanisms, or the production of the vector itself, as described in the following section.

#### 1.4.1. The VSV-G Pseudotyping

While HIV1 specifically targets CD4+ cells, the VSV-G protein was rapidly exploited to broaden the tropism of recombinant LV since this envelope binds membranes of most eukaryotic cells. Identification of its receptor, the low-density lipoproteins receptor (LDL-R) implicated in cholesterol cell homeostasis, only recently explained this wide tropism [86,87,88]. Many different tissues can be targeted with a VSV-G envelope, including neurons, liver, muscle, or even undifferentiated stem cells and embryos [7,33,89,90,91,92,93,94]. In the eye, the VSV-G envelope was used to target the retina (photoreceptors, retinal ganglion cells (RGCs), Müller cells, or retinal pigmented epithelium (RPE)), the cornea endothelium or the trabecular meshwork [38,95,96,97,98,99,100,101,102]. Coyle et al. modified the VSV-G envelope with PEGylation to decrease the complement inactivation of the recombinant vectors from human and mouse sera, which increased vector efficiency to target bone marrow and spleen [103]. Notably, adult subretinal injection of this PEGylated vector also increased lacZ transgene expression in the inner nuclear layer, but the type of targeted cells was not precisely described [103]. 

If the efficient targeting of RPE by VSV-G-pseudotyped LV is well recognized, photoreceptor transduction appears to be dependent on many different factors. The increased photoreceptor transduction observed after local damage at the site of injection [38,95] suggests that a physical barrier could reduce the efficiency of outernuclear targeting after LV subretinal injection compared to small AAV particles. Interestingly, age at injection and thus retinal development status influences the efficiency of HIV-1-derived vector to target photoreceptors. This indicates that this barrier could be built during development to form the mature retina and/or intrinsic cell factors that could limit transduction in adults [95]. The existence of a physical barrier was then challenged using enzymes to digest the extracellular matrix [104]. When subretinally injected, neuraminidase X, which targets sialic acid, increased the overall area of lentiviral transduction but not the density of transduced photoreceptors, which was still limited to the injection site. Similarly, the retina status, healthy or damaged, modifies the targeting of LV, also impacted by gliosis [98]. Lipinski’s study suggested that VSV-G targeting of photoreceptors could be species-dependent since a VSV-G pseudotyped vector could transduce cells of the outer nuclear layer (ONL) when incubated on humans but not on murine explants [100,105]. Likewise, some reports showed that the vector origin could have an impact on cell targeting since a VSV-G pseudotyped EIAV, unlike HIV-1, can transduce a significant amount of photoreceptors [38].

#### 1.4.2. Alternative Viral Envelopes

The Mokola virus envelope was also used in the retina to strongly target the RPE [106,107] but also Müller cells in degenerating conditions [98]. This work highlights once more how retinal conditions (healthy, degenerative, or stressed) can impact the cell targeting of the vectors. Another study showed that an LV pseudotyped with the Ross River virus (RRV) envelope mainly targeted RPE cells when expressing a CMV-GFP cassette [108], while it is more efficient to transduce Müller cells in vitro than a VSV-G-vector when expressing a cassette containing the CD44 promoter, which restricts expression in Müller cells [99]. However, this observation was not reproduced in vivo in wild-type rats, demonstrating again the importance of conducting vector-derived expression studies in the relevant model. Other envelopes, such as the Venezuelan equine encephalitis virus-derived glycoprotein (VEEV-G), also allow RPE transduction but present a higher toxic effect in mice, illustrated by decreased ERG response and ONL thickness with time [100].

Pseudotyping allows benefiting from the properties of particular envelopes, not only for specific cell targeting but also for transduction mechanisms. Murakami et al. increased the efficiency of transduction using SIV pseudotyped with the Sendai virus (SeV) hemagglutinin-neuraminidase (HN) and fusion (F) envelope proteins [109]. This vector can, with only a brief exposure time (a few minutes), efficiently transduce cells compared to the VSV-G envelope, which needs at least 24 h. This kind of development offers interesting opportunities to improve the efficacy and safety of vector application since the authors showed that a short time application of the vector is sufficient to achieve a therapeutic and long-lasting effect in the choroidal neovascularization (CNV) model. Therefore, they proposed that their procedure of injection, followed by the removal of the vector after five minutes, would greatly decrease the secondary effects of retinal detachment.

Similarly, the retrograde transport characteristic of the rabies-G envelope can be transmitted by pseudotyping. Rabies-G interacts with neuronal receptors, including neural cell adhesion molecule (NCAM) [110], nicotinic acetylcholine receptor [111,112,113], and the p75 low-affinity neurotrophin receptor (p75NTR) [114] and shows efficient targeting and retrograde transport in the brain [115]. In the retina, the Rabies-G-EIAV vector mainly targets RPE after subretinal injection but is less efficient than VSV-G pseudotyped EIAV in targeting anterior chamber tissue (endothelium and trabecular meshwork) [38]. Interestingly, as suggested for other pseudotyped vectors, the local damage at the site of injection also widens the vectors targeting other cell types such as ganglion cells and increases photoreceptor transduction [38,95,104]. Moreover, retrograde transport was also observed from brain to retina (RGCs, Müller cells, and amacrine cells) in the developing chicken after injection of these rabies-G EIAV vectors [116].

Pseudotyping can also be used to facilitate LV production. Indeed, as previously mentioned, the toxicity of VSV-G expression in HEK-293T limits the production of packaging cell lines (see Section 1.3.2). Therefore, researchers identified the baculovirus GP64 protein as an alternative envelope protein, which conserves a wide tropism because it functions as a fusion protein but can be safely and stably expressed for viral production [117,118]. Moreover, the fusion of GP64 with a complement decay factor (CD55) allowed it to escape from human and primate serum inactivation [119]. In the adult mouse retina, GP64-pseudotyped LV with a CMV promoter allowed expression in RPE but also in photoreceptors, despite it being a narrow area [108]. Strikingly, this targeting could be reproduced with the use of a Rhodopsin promoter [108]. 

#### 1.4.3. Chimeric Envelopes

Recently, to complement the use of native envelope proteins, chimeric glycoproteins were also engineered to develop new envelopes. Glycoproteins are composed of an ectodomain driving the cell targeting and a cytoplasmic tail required to generate viral particles. One strategy consists in preserving the VSV-G cytoplasmic tail, which is fused to a foreign ectodomain. When the Rabies-G ectodomain was used in such chimeric protein, LV production was improved [120,121]. However, replacing the ectodomain can also impair the internalization of viral particles. For this reason, co-pseudotyping with two glycoproteins was envisaged, one dedicated to the internalization and the other to the targeting, as demonstrated with the Singbis and the measle viruses (MV) [122,123,124]. Residues responsible for the binding of the native glycoproteins are first mutated, and the new ligand domain is fused to this ectodomain to be expressed outside the particles. Peptides such as single-chain antibody (scFv) or designed ankyrin repeat proteins (DARPins) can be used then to target specific cell types [125,126,127]. Interestingly, point mutations of the native MV glycoprotein also allow decreased neutralization by the human sera of the recombinant MV envelope compared to native MV [128]. This chimeric strategy was successfully developed for specific hematopoietic cell transduction but could be interesting in improving lentiviral cell targeting in the ocular field.

### 1.5. Biodistribution Safety for Eye Application

An undeniable advantage of the eye when considering LV-mediated gene therapy is the isolation of the organ from the rest of the body by the blood-retina barrier, which drastically reduces the risk of LV dissemination in non-ocular tissues. Nevertheless, it is essential to keep in mind that the ocular disease itself for which LV gene therapy is developed and/or the procedure of LV injection (Figure 3) can lead to Bruch’s membrane breach and then allow the passage of the LV into the bloodstream. While the immune system should inactivate the VSV-G pseudotyped vector, it is always important to validate the absence of LV distribution outside the ocular compartment.

In 2009, the short-term ophthalmic and systemic toxicity of a simian immunodeficiency virus from African green monkeys (SIVagm)-based lentiviral vectors carrying human pigment epithelium-derived factor (SIV-hPEDF) [75] was evaluated in non-human primate (NHP) retinas [81]. Reverse transcription polymerase chain reaction (RT-PCR) performed on urine, and serum samples showed no detection of vector sequence at day 1-8-30-90 post injection at the level of sensitivity of the assay. Likewise, PCR on whole blood samples did not allow the detection of any viral integration at the same time points.

Developed to treat the wet form of AMD, RetinoStat biosafety and biodistribution have been evaluated in 6-month studies, including 38 rabbits and 6 macaques [36]. A dose of 1.1 × 105 transducing units per eye (TU/eye) of this VSV-G pseudotyped, equine infectious anemia virus (EIAV)-based lentiviral vector, encoding for endostatin and angiostatin, was injected subretinally. Various fluids, tissues, and buffy coats were analyzed by quantitative polymerase chain reaction (qPCR) to determine the biodistribution, persistence, and shedding of the RetinoStat vector after subretinal delivery. No LV particles were detected above the lower limit of quantification (LLOQ) in rabbit fluids (urine, saliva, tear swabs, contralateral eye vitreous), and no dissemination was observed in blood or cerebrospinal fluid at any time point of the rabbit study. RetinoStat particles were observed within the sclera, a phenomenon explainable according to the authors by technical difficulties in ocular tissue isolation. Sporadic positive signals for the RetinoStat vector were observed only at early time points and in some collected tissues in rabbits but never above the LLOQ. The same observation was made in the macaque study.

This led to the development of a phase I clinical trial testing the safety of subretinal injection of RetinoStat (2.4  ×  10^4^ (*n*  =  3), 2.4  ×  10^5^ (*n*  =  3), or 8.0  ×  10^5^ transduction units (TU; *n*  =  15)) in patients with advanced neovascular age-related macular degeneration [129]. One of the six serious adverse events (AE) observed during the study was related to the procedure, but none of the serious AE or non-ocular AE were described as potentially linked to the LV itself. Regarding the shedding and integration of the LV, antibodies against RetinoStat were not detected in any patients, and once again, RetinoStat sequences were not detected in plasma, buffy coat, or urine after day 0 (only one patient presented some RNA sequence in plasma at day 0 but not thereafter).

The group that validated RetinoStat in rabbits and macaques (see above) also measured the safety and biodistribution of a VSV-G pseudotyped EIAV-based lentiviral vector carrying a coding sequence for the ABCA4 gene injected subretinally in rabbits and macaques, in a context of gene therapy for Stargardt disease [76]. The toxicity of the LV was measured after subretinal injection of a dose close to the maximal dose for each species (1.4 × 10^6^ TU/eye in rabbits, 4.7 × 10^5^ TU/eye in NHPs) 3 days, 1 week, 1–3–6 months post injection (p.i.) in 19 rabbits and 3 and 6 months p.i. in 6 NHPs. No death, body and organ weight change, tissue modification, or blood alteration was observed in treated versus control groups of both species. Regarding the shedding of the LV, despite the detection of the LV sequence by qPCR in two out of the six macaque liver samples, the level of detection was below the LLOQ. Moreover, LV sequences were also detected in the sclera and optic nerve, but the authors could not guarantee again that this was not just the result of imprecise dissection. Overall, this study demonstrated the safety of the LV and the localized expression within the targeted tissue. It provided support for the initiation of the first-in-man clinical trial of StarGen.

Following photoreceptor rescue of Usher type 1B syndrome mouse model treated with EIAV-CMV-MYO7A (UshStat), the safety of the vector was assessed in macaques [77]. Subretinal injection of 9.1 × 10^5^ TU/eye of UshStat did not induce death, differences in body/organs weights, blood chemistry, or blood cell counts compared to controls. These data support the safety of the LV and the clinical development of UshStat to treat Usher type 1B syndrome.

With the same idea to validate the safety of LV use for gene therapy clinical trials, but this time to treat RPE65-related Leber congenital amaurosis, we subretinally injected LV-RPE65 (2 doses tested: 2.8 × 10^5^ IU and 2.8 × 10^6^ IU in 100 μL) in healthy NHPs [74]. We evaluated the extraocular shedding of the vector by qRT-PCR targeting specific sequences of the LV, with a limited detection threshold of 10 copies of target matrix DNA per reaction solution. No viral sequence was detected in lachrymal fluid, serum, or urine at days 2-4-7-14-30-90 p.i. in any of the injected animals (*n* = 4). We also assessed extraocular genomic integration of the LV by qPCR targeting the transduced cassette on the genomic DNA of various organs. Neither eyelids, optic nerves, geniculate bodies, visual cortex, heart, liver, lungs, ovaries, kidneys, nor mandibular lymph nodes showed detectable level (10 copies/50 ng genomic DNA) of integrated lentiviral sequence.

Similarly, the recent study using an IDLV to transfer the CRISPR-Cas9 system targeting VEGFA in mice did not detect the qPCR recombinant viral genome in the liver, spleen, and testes of the injected mice [30]. Altogether, these studies confirm a safe use of subretinally injected LV regarding shedding and integration of the vector.

## 2. Development of Lentiviral Vectors for Ocular Therapeutic Applications

The main development of LV gene therapy in the posterior segment focused on diseases associated with RPE dysfunctions, including age-related macular degeneration and certain forms of inherited retinal dystrophy (IRD), since LV was shown to have a restricted tropism for the RPE probably due to its size and the outer limiting membrane (OLM) barrier [104]. In addition, RPE cells were also targeted to deliver genes coding for secreted or neuroprotective factors. In the anterior segment, LV was evaluated for gene transfer in the trabecular meshwork and in the cornea using both autonomous expression of transgene and secretion of surviving or anti-angiogenic factors. The LV gene transfer studies in the different ocular diseases are described below, and the type of diseases with their corresponding targeted locations are represented in Figure 4.

### 2.1. Gene Transfer Strategy for AMD

#### 2.1.1. Dry and Wet AMD

Age-related macular degeneration is characterized by progressive macula atrophy, the appearance of small yellow spot deposits called drusen, and a decline in central vision. The macula is mainly populated by cones with a unique presence in the fovea, essential for sight fixation and visual acuity. Deregulation of RPE function is thought to be at the origin of the disease, which then alters photoreceptor survival leading to a vicious circle. At the periphery of the macula, the loss of RPE cells generates a “geographic atrophy” typical of the dry AMD form, while the appearance of choroidal neovascularization (CNV) below the macula, provoking a separation between the RPE and the neuroretina is characteristic of the neovascular AMD also called exudative or wet AMD. The growth of neovessels deforms and then obscures central vision. Several angiogenic factors were then identified to be responsible for CNV growth, such as vascular endothelial growth factor (VEGF) and placenta growth factor (PlGF), leading to the development of efficient drugs to block and even regress CNV evolution [129]. For dry AMD, although several pathways were described to contribute to the disease progression (such as the inflammatory response and lipid homeostasis), no clear actors at the origin of the disease were identified to serve for effective therapeutic development [130].

#### 2.1.2. Gene Therapy for Wet AMD

As described above, CNVs in AMD depend on different angiogenic factors. Anti-angiogenic therapies have been successful now for 12 years, by delaying neovessel development for years and by efficiently protecting the retina [131]. However, these treatments require one intraocular injection every month or second month and are not well accepted over time. In addition, multiple injections increase the risk of endophthalmitis and hemorrhages. A long-term delivery is thus necessary to preserve the patient central vision for the rest of their lives. Although the gene therapy strategy appears to be a solution, full inhibition of VEGF is thought to be deleterious for the choroidal vessels and for certain retinal neuron survival. Indeed, VEGF supports the survival of different neurons [132], and the constant inhibition of VEGFA may lead to a loss of vessel fenestration and thinning of the vasculature [133]. In consequence, the ideal situation would be to block the pathological VEGF level without affecting the retina physiology. Some therapeutic candidates that were challenged using a lentiviral gene transfer approach are discussed below. For a more descriptive VEGF action and for an AAV vector review in this field, please refer to Koponen et al. 2021 [134].

#### 2.1.3. Anti-Angiogenic Strategy

Murakami et al. (2010) pseudotyped SIV vectors with a Sendai envelope (SeV-SIV) to optimize gene expression before testing them in two different approaches of gene transfer in the RPE cells to suppress laser photocoagulation-induced CNV [109]. While one SIV was produced to release the Fms-like tyrosine kinase-1 (sFlt-1) that receptor sequesters free VEGFA, the other vector coded for pigment epithelium-derived factor (PEDF). PEDF has both anti-angiogenic and neuroprotective actions [135,136]. Interestingly, in vitro studies showed that maximum gene expression with the SeV-SIV was already reached after 1 min, whereas a VSV-G-SIV reached maximal levels after 24 h. Subretinal injection of the vectors produced stable protein expression for both sFlt-1 and PEDF for at least 3 months. When these vectors were injected 2 weeks before laser impact, both proteins decreased the effect of laser injury by around 40%. The protective effect against CNV formation was even greater when the vector was injected 3 months before the laser injury. Interestingly, at 6 months p.i. the SeV-SIV-PEDF did not show side effects on the retina integrity, whereas the SeV-SIV-sFlt-1 produced a significant photoreceptor degeneration. In that case, choroidal vascularization was altered, as evidenced by indocyanine green angiography, but no alteration of the endothelial vessels was observed by electron microscopy. However, the choriocapillaris vessels contained malformed erythrocytes with contiguous thrombocytes. These experiments clearly show the advantage of using PEDF to prevent CNV and support photoreceptor survival. This study also reveals the delicate limits of decreasing the VEGF pathway to prevent CNV while maintaining retina integrity.

A lentiviral vector expressing a single-chain antibody directed against VEGF (scFv V65) was also tested to prevent CNV formation after laser injury in the mouse retina [137]. Following in vitro testing of the antibody release, the LV-EFs-V65 was injected 5 days prior to the laser injury. Two weeks after laser injury, a reduction of about 50% and 64% of CNV formation was observed in comparison to the no-vector-treated retina and LV-GFP-treated group, respectively. The LV-GFP-treated retina presented CNV invading the retina, a situation not seen after laser impact in the control retina. This suggests that probably a small inflammation due to vector delivery exacerbated the CNV development. Histological analyses also revealed complete preservation of the photoreceptor layers in the anti-VEGF group, whereas massive cell death was observed in the laser-lesioned site and the surrounding region in the two other groups. Although this vector served to dissect some VEGFA mechanisms in the RPE after light damage (see below Section 2.1.1), it appears to be an efficient tool to prevent neovascularization, even if the potential side effects remain to be carefully determined with long-term experiments.

Systems to modulate VEGFA expression were also used to prevent CNV formation. An LV vector was designed to establish the local expression of three miRNAs known to target Vegfa mRNAs. A multipurpose vector was generated by opposing two promoters, the phosphoglycerate kinase (PGK) ubiquitous promoter, driving the expression of eGFP to determine the transduction efficiency, and the RPE-specific vitelliform macular dystrophy 2 (VMD2) promoter, followed by an intron and the DsREd transgene, as a therapeutic cassette reporter. The intron can be replaced by any desired sequences, such as the miRNAs of interest here. Subretinal injection of such vector allowed specific expression of the transgene in the RPE cell population. The efficacy of the vector containing three miRNAs sequences integrated instead of the intron was validated in an in vitro model of endothelial cell tube formation dependent on VEGFA expression [138,139]. In vivo, the vector transfer performed 21 days before laser injury markedly prevented CNV formation. A 6-fold reduction in the CNV size was observed in the treated group compared to animals experienced with the control vector. These very promising data also require long-term study to assess the safety of this type of strategy.

A more drastic strategy consists of the knock-out of the VEGFA gene by editing approach. LV (HIV-derived) vectors were designed to express the Streptococcus pyrogenes (sp) Cas9 with GFP, which allows transduced cells isolation in order to quantify Indel generation [140]. After the selection of the most efficient sgRNA in vitro, an LV-U6-sgRNA-EFS-spCas9-eGFP (100 ng p24) was subretinally injected into the WT retina. The gene editing effect was quantified 35 days post injection. TIDE analysis revealed that the optimal sgRNA resulted in an efficiency of 83.6% of INDEL induction, with one base pair insertion representing 79.1% of the cases. After cloning and sequencing, 38.5% of the insertions (including those > 1 bp) provoked a knock-out of the Vegfa gene. Unfortunately, these experiments did not allow to have enough cells to estimate VEGFA expression. Although these results are very promising, further studies are needed to assess the effect of VEGFA levels in healthy and diseased retinas affected by neovascularization. Different factors, such as endostatin and angiostatin, were identified to counteract the action of pro-angiogenic factors and were thus interesting candidates for a gene therapy approach. An EIAV vector, the equine analog of LV, was engineered to drive the expression of each of these factors under the control of the VMD2 promoter specific for RPE cells [141]. Such vectors promote expression in the RPE of WT mouse retina as observed with a Lacz reporter, although the level of expression was around 6 to 10 times lower with the VMD2 promoter in comparison to the CMV one. Using the EIAV-VMD2p-LacZ vector, the expression was observed at least after one year and only in the RPE cells. Null vector (2 × 10^5^ TU) and EIAV-VMD2p-Endo (2 × 10^4^ TU) vectors injected 14 days prior to laser injury inducing-CNV and CNV were revealed 14 days later using perfusion of fluorescein-labeled dextran. The anti-angiogenic vector reduced by around 30% the volume of the CNV, whereas an EIAV-CMV-Endo provoked a more pronounced inhibition by more than 50%. Interestingly, a vector coding for endostatin and angiostatin separated by an IRES sequence, EIAV-VMD2p-Endo/Angio decreased by around 50% the CNV demonstrating that although the VMD2 promoter is much less active than CMV, a marked inhibition of the CNV can be obtained. No difference in the number of macrophages counted in the retina or the choroid was observed between the animals receiving the vehicle or the control or the potential therapeutic vectors.

These promising data resulted in the launch of a phase I/IIa clinical trial (NCT01301443) to explore the safety of the vector and identify potential benefits [142]. Following validation of the vector safety in rabbit eyes, 21 patients with advanced neovascular AMD were enrolled and injected with increasing vector doses. The three first groups of patients had a low visual acuity with a mean of 21.4 Early Treatment Diabetic Retinopathy Study (ETDRS) letter score (corresponding to 20/400 Snellen grad) and received an injection of 2.4 × 10^4^ (*n* = 3), 2.4 × 10^5^ (*n* = 3), and 8 × 10^5^ TU (*n* = 3). Regarding drug delivery, the needle was in contact with the retina in the first attempt to facilitate the vector penetration into the retina. Because of leakage in the vitreous, a subretinal route by retinotomy was then chosen. A 41-gauge needle was used to first rapidly inject 300 µL of a balanced salt solution to form a bleb. Then the needle was replaced by another one to inject the vector in the space generated. A minimum interval of 14 days was applied between individual patients injected. One serious (macular hole) and two non-serious adverse events were reported (peripheral tears), mainly due to the operative procedure rather than the vector as stated by the authors. All the events resorbed rapidly, and only a transient mild inflammation was observed after the vector administration, suggesting a suitable tolerance of the vector.

With the two higher vector doses, the maximal levels of ANGIOSTATIN and ENDOSTATIN in the aqueous humor were reached at around 5 weeks p.i. and remained at a plateau during the 48 weeks of monitoring, revealing the long-term ability of the vector to stably express the transgene. Persistent gene expression was also detected in one patient after 4.5 years. A 10- and > 300-fold increase in expression level was observed for ANGIOSTATIN and ENDOSTATIN, respectively.

No obvious or discrete changes were observed in images taken by optical coherence tomography (OCT), with the exception of one patient (out of 21) showing the disappearance of macular edema and recovery of retina thickness. In addition, seven patients of this cohort still received anti-VEGF injections as a therapeutic response to the disease development, suggesting that even high levels of these transgenes failed to modify advanced neovascular AMD evolution. The present clinical trial involved many patients who had stopped anti-VEGF treatment because of a lack of visual benefit. In this situation, it was difficult to expect clear ameliorations with ANGIOSTATIN and ENDOSTATIN treatments. Further tests in AMD patients with less severe situations should reveal whether this vector may have therapeutic potential.

### 2.2. Gene Transfer for Retinopathy of Prematurity

Oxygen supplementation for preterm children generates a hypoxia-like situation once the child returns to a normal oxygen level. In this context, a delayed retinal vasculature develops, as well as aberrant vascularization that may lead to retinal detachment and invasion of neovessels into the vitreous. This retinopathy of prematurity (ROP) is a leading cause of childhood vision loss and blindness worldwide [143]. The main challenge is to prevent pathological angiogenesis, currently attempted by using repetitive injections of anti-angiogenic factors. A proposed alternative is a gene transfer approach to better control neovascularization.

The lncRNA taurine upregulated gene-1 (TUG1), upregulated during hypoxic conditions [144], was shown to be involved in pathological angiogenesis [144]. One action of the lncRNAs is to buffer specific miRNAs. To explore the possible role of Tug1 in an ROP mouse model, oxygen-induced retinopathy (OIR) was triggered by exposing PN7 mice to a hyperoxic condition (75% oxygen) for 5 days, followed by a return to a normoxic environment [144]. This protocol induced a marked alteration of the retinal vessel plexus organization with avascular zones and multiple foci of neovascularization, as well as significant upregulation of Tug1. Mice received intravitreal injections of either LV (0.5 µL) coding for Tug1 shRNA (3 × 10^8^ TU/mL) or a control vector. The therapeutic LV prevented Tug1 induction and maintained a normal level of mmu-miRNA-299-3p, one of the lncTug1 targets. The effect was also evidenced in the retina by noticeably preventing the formation of neovessels and allowing a better vasculature plexus formation. In addition, the massive cell death observed in the control OIR-treated animals was almost completely abolished by the LV-Tug1-shRNA, and the inflammatory factors such as Il-6, Il-1β, and TNF-α were decreased in accordance with this observation. Finally, Tug1 inhibition prevented the marked upregulation of VEGF, with only a few little foci of expression in the retina.

Another approach to inhibit the VEGF pathway is to target VEGF receptor 2 (VEGFR2). The advantage of such a strategy is to maintain the VEGF role in normal vasculature development and in certain neuron survival [145]. To that aim, LV coding for shRNAs against Vegfr2 or Stat3 mRNAs, under the control of the VE-Cadherin promoter, was produced and first validated to downregulate VEGFR2 signaling [145] before being tested in an OIR rat model. OIR started in rats 6 h after birth by exposing them to low (10%) and high (50%) oxygen in the 24 h/24 h cycle. At PN8, LV subretinal injection (1 µL of 1.0 × 10^9^ viral particles per ml) was performed, and animals were placed in a 50% oxygen environment until PN14, when they were transferred into normoxic conditions. Such conditioning induced intravitreal neovascularization (IVNV) and zones of atrophy in the retinal vasculature. At PN20, a 32% decrease in IVNV and an approximately 20% thicker retina was observed in the LV-Vegfr2shRNA-treated group in comparison to controls (LV-LucshRNA). A significant reduction (18%) of atrophic areas was also observed. Retinal activity assessment of the LV-Vegfr2shRNA-treated group showed a slightly better response to some stimuli as recorded by electroretinogram. VEGFR2 knockdown had no effect on circulating VEGF nor on animal growth.

Within the same OIR model, the effect of VEGFA knockdown in the Müller cells was investigated with a LV expressing a shRNA only in these cells thanks to the CD44 promoter. LV-CD44p-VEGFAshRNA-GFP specificity and efficiency in inhibiting VEGFA in Müller cells and preventing IVNV were first evaluated [146] and then compared to LV-CD44p-VEGF164shRNA-GFP. Lentiviral vectors (LV-CD44p-lucifshRNA-GFP in the control group) were subretinally injected at PN8, starting point of OIR [147]. Only retinas presenting GFP in the Müller cells were taken into account. At PN18, LV-CD44p-VEGFAshRNA-GFP and LV-CD44p-VEGF164shRNA-GFP decreased IVNV by more than 30%, while no change in avascular retinal areas was observed. A significant thinning of the ONL was observed in all injected groups, including the one with PBS, suggesting that it might have been caused by the retinal detachment induced by the injection rather than due to a toxic effect of the vectors. Since VEGF inhibition may induce retinal cell death [148,149], retinal activity was monitored with both focal ERG to study only the transduced area and full-field ERG to measure the whole retina activity. Interestingly, no difference between therapeutic and control vectors was observed in focal ERG, whereas full-field ERG revealed a modestly improved response in retinas treated with LV-CD44p-VEGFAshRNA-GFP or LV-CD44p-VEGF164shRNA-GFP. These experiments showed that the inhibition of VEGFA or VEGF164 had no adverse effects and even seemed to have a protective effect during OIR. To unravel the involved mechanism, different neurotrophic factors known to be linked with VEGF activation were measured. At PN18, no significant differences were observed between groups for the levels of EPO, NT3, GDNF, NGF, and brain-derived neurotrophic factor (BDNF), whereas all these factors were more expressed at PN32 in the LV-CD44p-VEGFAshRNA-GFP group only. Because the VEGF164 variant is often associated with inflammation and pathological conditions [148], targeting this variant may be beneficial for ROP treatment by modifying the disease course evolution without altering VEGFA-dependent retinal development.

The use of LV helped to dissect the mechanisms involved in the process of ROP by identifying different actors as described above. Although some approaches are promising, such as the targeting of Tug1, it is somehow difficult to justify treatment for children that can be active the whole life when targeting only a limited time period is sufficient. The exploration of transient expression or inducible promoters [150] controlling the level and time of transgene expression would undoubtedly improve vector safety in the long term and could help for a clinical application.

### 2.3. Gene Therapy for Inherited Retinal Dystrophies (IRD)

Many gene mutations are at the origin of IRD affecting rod, cone, or RPE functions. More than 400 altered genes were identified to be causative for IRD, which englobe Leber congenital amaurosis, retinitis pigmentosa, night blindness, Usher syndrome, Stargardt disease, cone-rod dystrophies, Bardet-Biedl syndrome, and several others [151]. In the case of retinal diseases altering the first rod function, the photoreceptors degenerate in the retinal periphery, and the degenerative process progressively reaches the central region enriched in cones, which degenerates secondly to rod loss. For such rod-cone dystrophies, the patients often have night blindness evolving into a tunnel vision that can be followed by total blindness, whereas for cone-rod dystrophies, central vision and color perception are altered first, and patients feel discomfort with bright light. No treatment exists for these diseases, with the exception of Leber congenital amaurosis type 2, for which AAV-derived gene therapy is commercially available (Luxturna™, Philadelphia, PA, USA). The accomplishment of such treatment paved the road for ocular gene therapy, demonstrating the pertinence of a gene augmentation approach for recessive diseases.

#### 2.3.1. IRD and Gene Replacement or Correction Strategy

##### Retinitis Pigmentosa

The first IRD mouse model investigated to test LV efficacy was the Rd1 mouse, which has a mutation in the Pde6b gene leading to phototransduction alteration in rods, rod death, and successive loss of cones. Young animals were subretinally injected at PN2 to 5 with LV-Rhop-GFP or LV-Rhop-Pde6b (around 5 × 10^5^ TU) [152]. The Rhodopsin promoter (Rhop) was used to restrict the expression to rod photoreceptor cells. Rd1 mice treated with LV-Rhop-Pde6b presented 2–3 rows of photoreceptors at 6 weeks of age when the control group had no more sensory cells in the ONL. Opsin-positive photoreceptors were still detected until 24 weeks of age. These experiments demonstrated the long-term effect of the vector to maintain the survival of photoreceptor cells when injected peri-natally. However, since only a few cells were targeted, improved procedures are necessary to reach the photoreceptors when the retina is fully formed [104].

##### Leber Congenital Amaurosis


*Gene Replacement*


A more promising target to be explored for an ocular gene replacement strategy was the RPE cells to augment gene expression in RPE65 deficiency. An LV was constructed with the RPE65 promoter short sequence driving the expression of hRPE65. Perinatal treatment of the Rpe65^−/−^ mice markedly protected against cone degeneration and restored their function. Interestingly, cone protection was also observed in the nearby region of transgene expression, which is mainly recognized to be restricted to the bleb formed during the injection. A linear relationship between the treated area and the surrounding rescued area can be determined and may serve in the future to treat the macula without the need for macular detachment to avoid damage due to the injection [153]. In another mouse model of RPE65 deficiency in which the R91W missense mutation leads to a slow degeneration, gene augmentation was also efficient during the degenerative process [154]. Interestingly, gene therapy treatment of mice at one month of age, when only 36% of the cones are expressing cone markers, rejuvenated many cones. Indeed, 3 months after gene therapy, 64% of the cones expressed the GNAT2 cone marker. These experiments show the LV’s high efficacy for a gene replacement strategy in the RPE cells and the possibility of counteracting the degenerative process in the RPE65 deficiency, an action not reported to date with other vectors.

A GMP-like production of the LV-hRPE65 was then tested in monkey (*Macaca fascicularis*) eyes to evaluate the safety of the vector [74]. Eye monitoring was performed at different time points to evaluate retina integrity and retina reattachment kinetic after vector delivery. Animals were injected with two different vector doses (2.8 × 10^5^ IU and 2.8 × 10^6^ IU in 100 µL TSSM per animal, *n* = 2 per dose) or with vehicle (TSSM) and received a single topical application of dexamethasone/oxytetracycline ointment but no anti-inflammatory agents prior to injection. The possible shedding of the vector in different tissues and organs was quantified. For all the samples tested (*n* = 91), none was above the detection threshold, revealing a restricted localization of the vector after its injection.

The eyes injected with the vectors showed a delay in the retina reattachment. In one animal, leakage of the high dose vector was observed, provoking an intense inflammation that was controlled by daily intramuscular administration of methylprednisolone (1 mg/kg) for 3 days. The inflammation resorbed after one week. A vasculitis-like reaction was observed for all LV-RPE65-treated eyes with a perivenular whitening, suggestive of frost-branch angiitis, which resolved after 14 days. This reaction is normally observed in the eyes after a viral infection. A small thinning of the ONL thickness was observed in all groups, with a more pronounced one in two cases, revealing the deleterious effect of subretinal injections. These results show a limited tolerance for LV after the subretinal administration route and suggest the transient need for anti-inflammatory drugs to counteract the inflammation side effects.

Finally, these preclinical data were reinforced by the demonstration that LV can increase the level of RPE65 mRNA in human iPSCs-derived RPE cells [155], strongly supporting the potential of this vector for clinical applications of gene expression in the RPE.


*Gene Editing*


New editing strategies, based on modified Cas9, allow for specifically replacing a cytosine or an adenine without inducing DNA double-strand break [156]. An adenine base editor (ABE, consisting of a Cas9 without DNAse activity fused to an adenosine deaminase) carried by an LV was used to correct a Rpe65 gene mutation in the Rd12 mouse model, which has a stop codon imposed by an adenine mutation [157]. Such an approach was used because a homology-directed repair (HDR) strategy was shown to be poorly efficient, with only 1% of corrections and Indel formations [158]. In the context of the ABE approach, since the Rd12 mouse does not contain an NGG protospacer-adjacent motif (PAM) sequence close to the mutation, other Cas9 were screened to recognize different PAM sequences, such as the xCas(3.7). After identifying the best sgRNA and codon optimization of the ABE, an LV-U6-sgRNA-CMV-ABEmax was generated and subretinally injected (1 × 10^6^ TU per injection, co-injected with 5 × 10^7^ TU LV-CMV-eGFP to follow the expression) in PN28 Rd12 mice and the eyes were analyzed 5 weeks later. Two gRNAs were tested and compared to the control vector expressing ABE and a non-targeting (NT) gRNA. One construct allowed the restoration of RPE65 expression in 32% of RPE cells of the treated area, which correlates with the 29% of gene correction assessed by deep sequencing. Moreover, Indel mutations corresponded to only 0.48% of cases demonstrating the very suitable performance of this vector and its safety characteristic. The retina activity, attested by ERG, was reestablished to a level of around 65% of the WT (for the b-wave amplitude), while no signal was recorded from the control group. In low-light situations (1 lux), the visual acuity of the LV-U6-sgRNA-CMV-ABEmax injected eyes was very close to the WT animals, whereas no behavioral responses were observed in controls. These data are in accordance with electrical activity recorded in the primary visual cortex (V1), despite a significant small delay in the response. This work demonstrates the feasibility of one single vector to efficiently correct gene mutation in a substantial number of RPE cells allowing for the restoration of visual functions almost similar to a healthy subject. This was clearly proven for rod function, and it would be valuable to have complementary results on cone function. Nonetheless, this technology opens the way for gene corrections in several diseases affecting RPE cells.

##### Usher Syndrome

One main advantage of the lentiviral vectors is their ability to cargo long sequences of up to 9 kb, rendering them attractive for certain diseases such as Usher syndrome type-1b (Ush1b), which affects both hearing and vision as well as the vestibular system in certain cases. The shaker1 mouse model partially recapitulates the disease with congenital deafness appearance and vestibular dysfunction due to mutations in the Myo7a gene [159]. However, despite the identification of biochemical alterations, no retinal degeneration was documented, nor obvious ERG response modifications, but some biochemical alterations were identified.

The first proof of principle that LV can efficiently transfer a large transgene into the retina was tested in the shaker1 mouse after in vitro validation of transgene expression and biological efficacy. Due to the limited mouse retina phenotype, the researchers focused on the biochemical characteristics of RPE and photoreceptors. Interestingly, melanosomes in cultured shaker1 RPE migrated in the cells with detours, whereas in the WT RPE, melanosomes followed a straightforward path [160]. MYO7A gene transfer with an HIV-derived vector (LV) reestablished melanosome motility in the mutant RPE cells. Consistently, in studies of in vivo LV-MYO7A subretinal injection (1 × 10^7^ TU/mL) at PN1, electron microscopy analyses at PN16 revealed that gene transfer restored the correct distribution of melanosomes in the apical region of the RPE cells. Moreover, while opsin accumulation was attested by immunogold in the mutant mice, photoreceptors rescued by the LV treatment presented a similar number of immunogold particles in the connecting cilium to the control group.

In another study [77], newborn shaker1 mice were subretinally injected with a recombinant EIAV vector coding for hMYO7A cDNA named UshStat. The transgene activity, which can be detected by antibodies specific to the human protein, was detected in RPE and photoreceptor cells. Four weeks p.i., animals were dark adapted and then submitted to a 200 lux stimulation for 10 min. While in the non-transduced area, the protein remained in the outer segment compartment, the α-transducin translocated normally in the transduced area, from the outer segment toward the inner segment, showing repair of some transport processes through the connecting cilium.

Although the shaker1 mice do not present a spontaneous retinal degeneration, photoreceptor loss can be induced by continuous light exposure to 2000 lux for 6 days. In such conditions, the mutant mice lose around 40% of photoreceptors in comparison to the WT retina. The injection of the UshStat vector prevented the decrease by around 50%. Such a result is encouraging and is in line with the observed vector transduction efficacy of 53% at maximum.

Safety studies in monkeys revealed that the vector dose of 9.1 × 10^5^ TU/eye, i.e., the maximum dose tolerated, induced the production of human MYO7A protein in RPE cells and photoreceptor segments [77]. No change in blood cell composition was observed. The retina bleb, formed after subretinal injection, was present in all animals. Vitreal opacity due to inflammatory cells was present in all EIAV vector-treated eyes but resolved at the end of the study (3 months). In one animal, the vector leaked into the vitreous and provoked a stronger inflammation. The authors did not indicate whether glucocorticoids or other anti-inflammatory drugs were given pre and post injection. No eye fundus nor OCT imaging were provided to show the retina integrity maintenance after the procedure.

These data highlight the potential of the EIAV vector for retina gene delivery, even though the treatment appears feasible mainly (or uniquely) during the perinatal period when the OLM is not formed and does not prevent the vector entry into the neuroretina [104]. Anti-inflammatory drugs appear to be necessary to avoid potential alteration. The fact that the bleb remained in the monkey retina suggests the induction of such deleterious effects since, in normal conditions, a reattachment is expected following the next two days [74].

A phase I/IIa clinical trial was performed with UshStat and achieved in 2019. However, no peer-reviewed publication to date relates the study observations (https://clinicaltrials.gov/ct2/show/NCT01505062, accessed on 25 July 2022).

##### Stargardt Disease

Stargardt disease is the most common hereditary disease affecting the macula, responsible for high visual acuity. Although the disease onset is very frequent during the juvenile period, the disease evolution is very variable and can take decades to provoke the full atrophy of the fovea (located in the center of the macula). The disease is due to mutations in the large gene ABCA4 transporter, involved in the transport of *trans*-retinal metabolites necessary for the recycling of the visual pigment. The EIAV vector was again chosen because of its large cargo sequence capacity to explore the feasibility of restoring gene function in Abca4-deficient mice [161]. Because rodents do not have a macula, the retina phenotype of the mutant is subtle. Alteration of all *trans*-retinal metabolism results, for instance, in the accumulation of the A2E lipofuscin compound, which presents a strong fluorescence making it easily detectable. Subretinal injection of EIAV-CMV-LacZ into mouse WT retina at PN4-5 (5.0 × 10^5^ TU) allowed the transduction of around 5% of photoreceptors as well as RPE and some Müller cells in the treated area [161]. A similar delivery of the EIAV-bRho-LacZ vector resulted in limited expression of the transgene in photoreceptors as expected, with a higher rate of specificity for the sensory cells. The authors stated that around 20% of these cells were transduced, but the representative picture suggests transduction below 10%. The X-gal staining is not the best method to quantify cell specificity, and it is unfortunate that no antibody against LacZ was used. Nonetheless, injection of EIAV-CMV-ABCA4 or EIAV-Rho-ABCA4 vectors markedly reduced (by at least three times) and with the same potency, the level of A2E, 6 and 12 months post injection. These data show that even a small percentage of transduced cells may have a strong effect on certain cell metabolism processes.

The safety of EIAV-CMV-ABCA4 or GFP vectors were tested in monkey (*Macaca mulatta*-1.4 × 10^6^ TU/eye) and rabbits (4.7 × 10^5^ TU/eye) (StarGen, [76]). The dose difference is explained by the tolerance of rabbit eyes to higher vector doses than monkey eyes. Before the injection, a 41-gauge needle, connected to a tube, is filled successively with PBS, an air bubble, and finally 100 µL of the vector. This allows precise injection of the vector only. Animals were injected and followed for up to 3 months p.i. A transient inflammation was observed with the StarGen vector, but no elevation of the intraocular pressure was present. The ERG responses were also not affected. Concerning gene expression, GFP was detected in RPE cells and photoreceptors, as well as several interneurons and glial cells composing the inner nuclear layer. The percentage of transduced cells was not provided and is difficult to assess because of picture overexposition. No alteration of white and blood cell numbers was observed nor blood chemistry. The retina integrity following the bleb formation was monitored and reflected the retina detachment. Slit lamp, indirect ophthalmoscope and fundus examination were performed. A slight elevation of the retina due to the bleb was observed in 41% of the monkeys in all treated groups with resorption after 15 days, which is a longer period than usually observed. OCT imaging could have been informative in evaluating a possible loss of retinal cells after this time period. The effect of retina elevation in both treatment conditions (vector and vehicle) was more pronounced with the vector in the rabbit eye with a higher frequency of affected animals (85–87%) and a longer duration (day 22 and day 85 for two animals treated with the buffer).

Vector distribution was analyzed at different time points post injection. The vector was detected by RT-qPCR exclusively in the visual system, including the optic nerve and the optic chiasma for one animal. No vector was identified in the brain, gonads, liver, and spleen. Intriguingly, the copy number decreased in the retina/choroid tissues from 0.2 vector copy per cell at 3 and 8 days p.i. to 0.02 and 0.003 at 29 and 92 days post injection. A quite similar pattern, but starting with fewer copy numbers per cell (0.02), was observed in the sclera. The authors suggested that this decrease may be due to the presence of vectors within the bleb, which resolved with time. However, this does not explain whether the vectors diffuse elsewhere or are degraded. Because no vectors were detected in other organs, the hypothesis of degradation is the most likely. Nonetheless, the main finding of this study was the advantage of the vectors remaining within the targeted organ and not significantly transducing vital organs. Histological analyses evidenced that, in both rabbits and non-human primates, retinal degeneration of the detached retina occurred in the control and vector-treated groups, even if the severity of retina alteration was slightly higher in the last one.

A phase I/IIa clinical trial was launched with StarGen in 2011 (SAR422459, NCT01367444) and completed in 2019 [162]. The goal of this clinical trial was to assess the safety of the vector (primary endpoint) and potential improvements in vision and structural changes. This results in a three-year study involving 22 patients who received subretinal injections of the EIAV-ABCA4 vector at different doses (1.8 × 10^5^, 6 × 10^5^, and 1.8 × 10^6^ TU) through a retinotomy procedure located in the temporal region of the optic nerve and anterior to the major superior vascular arcade. The worst eye only was treated. The bleb diffused under the foveal region in 12 patients, whereas in 10 it remained in the extra-foveal region. All patients were treated with topical corticosteroids for 2 to 6 weeks. In total, nine patients received supplementary corticosteroids (intravenously or periocular). During the 3-year study, 183 adverse events were reported, among which 18 related to product administration (including the surgery procedure). The adverse events were conjunctival hemorrhage (*n* = 11), intraocular pressure (IOP) increase (*n* = 5), or ocular pain (*n* = 6). One serious adverse event supposed to be linked to the surgery was reported: the IOP increased from 18 to 35 mm Hg 1 day after the injection. This was considered severe because it lasted until week 43 with a 31 mm Hg IOP, even under medication. A miscarriage 2 years after the treatment was also recorded as a serious adverse event, but the patient could give birth safely one year after. Vector shedding in blood and urine did not lead to the detection of quantifiable vector particles. Immune reaction against vector proteins or the transgene product (VSV-G2 envelope protein, neomycin phosphotransferase, p26 protein, and ABCA4 protein) was observed in three patients for ABCA4 and in two patients for VSV-G at week 24.

Concerning best-corrected visual acuity (BCVA), it was first established with repetitive examinations that for this population, a gain is considered as it when more than eight letters of improvement were recorded [163]. No significant amelioration was observed, with the exception of one patient showing a sustained increase in BCVA. Nevertheless, since the non-treated eye improved similarly, the authors suggest that the examination prior to treatment was probably unconsciously biased. No sustained and significant clinical improvement for static and kinetic perimetry was demonstrated. Color fundus photography and fundus autofluorescence (FAF) imaging revealed in one patient, injected with the highest dose, a change in the hyperreflectivity of macular flecks with a significant attenuation with time. This reflectivity diminution cannot be due to a change of RPE atrophy as attested by spectral domain-OCT (SD-OCT) imaging. However, enlargements of hypoautofluorescent areas were observed in the treated area of six patients and not in the contralateral uninjected eye suggesting a progression of the disease. Retina activity was monitored only with a 30 Hz stimuli to obtain a reliable signal. No significant differences were observed between untreated and treated eyes.

This clinical trial first revealed the difficulty of monitoring the efficacy of treatment in patients with low vision, difficulty of fixation, and advanced stage of the disease corresponding to multiple changes in the retina appearance. Moreover, the retina fragility in these affected conditions renders it difficult to determine the origin of some deleterious changes after treatment, such as the hypoautofluorescent area progression observed in a high rate of patients. It is not clear whether the retina detachment induced by the injection is responsible for such progression or if this change is solely due to some inflammation and immune reaction provoked by the vector. The macular flecks disappearance in one patient may reflect the potential action of the transgene but could not be associated with other clinical features within this study.

All these experiments demonstrate the EIAV vector’s ability to produce large transgene expression in targeted cells. Nonetheless, the drug delivery procedure still has to be improved regarding the surgical approach and by favoring the tropism of the vector for photoreceptors to optimize gene transfer efficacy and safety. Clinical studies revealed that the EIAV vector is, in general, well tolerated, but the alterations observed by autofluorescence in almost one-third of patients suggest further investigation of the vector safety.

#### 2.3.2. IRD and Neuroprotective Strategy

As stated above, many different mutated genes are at the origin of IRDs, leading to photoreceptor dysfunctions and death. For rod dystrophies, the peripheral retina starts to degenerate with a slow progression toward the central retina, leading to tunnel vision. Depending on rod loss severity, cones degenerate secondly due to a lack of adapted metabolism and trophic support [164,165,166]. Many diseases are not treatable so far due to the transgene size limit of vectors or because of a need for models. In addition, for certain diseases, a gene replacement treatment at a late disease stage did not prevent retinal degeneration, although the visual function was ameliorated [167]. In consequence, a global treatment to support photoreceptor survival is required.

To test a neuroprotective approach by gene therapy, two different rodent models were chosen. The first model presents a deficiency, due to a mutation in the Mertk gene, of RPE function to phagocyte photoreceptor outer segment debris (each photoreceptor continuously loses outer segment fragments), leading to photoreceptor debris accumulation and progressive photoreceptor death [168,169]. The second model is the Rds mouse, which presents a defect in the structural protein peripherin [170,171]. Two SIV vectors were produced to prevent photoreceptor cell death in these models, one coding for the neurotrophic PEDF and the other one for the fibroblast growth factor 2 (FGF2). SIV-hPEDF and SIV-hFGF2 (10 µL of 2.5 × 10^7^ TU/mL) were subretinally injected separately or simultaneously [79]. The trophic factors were detected in the retina at a concentration of around 2–3 ng/mg of protein 4 weeks after gene transfer. For therapeutic evaluation, 3-week-old RCS rats were injected and analyzed 8 weeks later. Treated retina with SIV-hPEDF or SIV-hFGF2 had around 100% and 50% more photoreceptors in the injected area, respectively, in comparison to the SIV-EGFP group. Co-injection of both vectors increased by 2- to 3-fold the number of protected cells and slightly extended the protected region. Interestingly, in this condition, ERG response was markedly improved. In Rds mice at 4 weeks post treatment, all single or dual vector injections showed a similar benefit on ERG response, with a b-wave more than twice higher compared to the control group. However, the authors concluded that no morphological rescue was observed for the number of photoreceptors remaining in the retina, but they did not analyze the integrity of the outer segments, known to be altered in these mice.

The RCS rat is recognized to respond well to neurotrophic factors, and this model served to validate the gene transfer by SIV vectors. Whether PEDF and FGF-2 may have an interest in neuroprotection in other models of IRD remains to be determined.

### 2.4. Gene Therapy Strategies for Glaucoma

Glaucoma is characterized by an elevated IOP and the loss of RGC function and survival. RGCs form the optic nerve that connects to the brain, explaining how their degeneration in glaucoma inexorably condemns any visual perception. The major cause of the elevated IOP is a deficit of aqueous humor absorption from the anterior chamber by the trabecular meshwork in the Schlemm’s canal. The actin cytoskeleton of the trabecular meshwork participates in the formation of aqueous humor drainage resistance. Many drugs can help to regulate intraocular pressure (IOP) through topical applications. However, some patients are resistant to these drugs, and some RGCs are particularly sensitive to high-pressure environments. In consequence, the development of gene therapy for glaucoma either aims to target cells of the trabecular meshwork to improve fluid reabsorption or to sustain RGC survival.

#### 2.4.1. Targeting the Trabecular Meshwork

The first evidence that LV can transduce trabecular meshwork was obtained using an FIV vector in a human eye organ culture [101]. The same group then presented the proof of concept that these cells can be efficiently targeted in vivo in cat eyes with 107 or 108 TU depending on the construct [102].

To render the trabecular meshwork more permeable for fluid drainage, the cytoskeleton organization was targeted by different approaches. One study focused on the Exoenzyme C3 transferase (C3), a Rho inhibitor that can modify cell morphology by cytoskeleton disorganization [172]. The first proof of principle was made using adenoviral vectors in cultured cells of the trabecular meshwork to show a transgene effect on these cells. This vector also increased the flow in perfused non-human primates and human eyes. These experiments demonstrated that C3 is a very interesting candidate gene to control aqueous humor drainage from the Schlemm’s canal. Similar data were obtained with perfused macaca fascicularis eyes treated with an FIV-C3 vector [173]. Then, an FIV lentiviral vector FIV-CMV-C3-GFP (FIV-C3.GFP) was constructed, and its efficiency was validated in human trabecular meshwork cells. The impact of this vector on actin disorganization led to cell morphology changes toward round-shaped cells and spaced out the cells. The intracameral (in the anterior chamber) injection of 4 × 10^6^ TU of FIV-C3-GFP in the rat eye led to a marked transduction of cells within the Schlemm’s canal region. Inflammation and side effects were informed only 48 days p.i. Macro-observations did not reveal obvious changes in the anterior chamber [82]. GFP expression peaked at 21 days p.i. The IOP significantly decreased as soon as 3 days p.i. but progressively returned to normal level when GFP expression dropped.

The same authors then brought this technology into an animal model closer to the human eye physiology by testing it in *macaca mullata* (Rhesus monkey). The trabecular meshwork was transduced with LV-C3-GFP or LV-GFP vectors [83]. A high expression was detected in vivo during 113 days with LV-GFP and to a lesser extent with the LV-C3-GFP (2.5 × 10^7^ TU for both vectors). Only one eye out of eight presented a mild inflammatory reaction (cornea edema and flare) on day 3 post injection with no signs anymore on day 7. No other side effects were reported. The LV-C3-GFP vector induced a decrease in IOP, whereas the control vector did not, with a maximal effect at 3 days p.i., but the difference with the control group reduced with time to reach a comparable level at day 119. Interestingly, maximal detection of GFP was observed when the IOP was low, followed by GFP expression decrease during the reestablishment of normal IOP, as previously described in rat eye [82]. Trabecular meshwork was then analyzed by histology at 70 days p.i., when the IOP was significantly decreased. Both reduced cellularity and stroma loss were observed in the justacanalicular tissues of the trabecular meshwork, while reduced cellularity was also observed in the inner wall of the Schlemm’s canal, only in the LV-C3-GFP-treated group.

Although the vector safety appears adequate, and the C3 gene is shown as a promising transgene for the LV approach to reduce IOP, these data do not support the long-term effect of the gene expression. GFP monitoring indicates that it is a problem of promoter choice or targeted cells rather than transgene. It would be interesting to verify if the CMV promoter was silenced over time and whether the vector was still present in the Schlemm’s canal by the RNAscope approach, for instance. If the vector was not present anymore, this may indicate that the transduced cells were lost. Previous studies have shown proliferating cells in the trabecular meshwork suggesting a renewal capacity of these cells [174]. To allow the maintenance of the transgene in this tissue, LV-C3 should target the trabecular meshwork stem cells [175]. This may then have an impact on the Schlemm’s canal cell organization. Nonetheless, long-term transduction was already documented with the FIV vector in cats (2.3 years [84]) and monkeys (1.5 years [176]), suggesting that the injection procedure can be determinant as well.

A similar approach was performed using an LV coding for miR-146a [177], expressed during replicative senescence of human trabecular meshwork cells. MiR-146a was shown to repress several genes associated with inflammation and senescence and reduce intracellular reactive species [178]. MiR-146a is upregulated when trabecular meshwork cells are submitted to stress [179]. In vitro in mechanically stressed trabecular meshwork cells, LV expressing miR-146a (LV-146) decreased the expression of some genes regulating inflammation, such as IL8, IRAK1, and COX1 [177]. Intracameral injection of 25 µL of LV-146 (5.8 × 10^8^ pfu/mL) led to a rapidly detectable reduction in IOP, but the IOP difference with the control group started to diminish only after 3.5 months. This is probably due to the high volume to inject regarding the low titer preparation, with a significant loss of the solution, leading to reduced vector availability. Nonetheless, a more sustained effect was obtained using a higher dose of vectors (1.0 × 10^9^ pfu/mL) with a mean difference of about −6 to −4 mmHg after 10 months. Visual acuity characterization at 7 months p.i. revealed that the vector delivery did not alter visual function. A 4-fold increase for miRNA-146a was detected in the treated region, with no change for TNFa, IL1b, and CD68 genes.

These experiments clearly show that the long-term expression of a transgene can be achieved in the trabecular meshwork by LV delivery without side effects. In addition, miRNAs also appear to be an interesting tool to control the metabolism of these cells in order to ensure proper functions.

Another approach to decrease IOP consisted of developing an HIV-based gene transfer to increase the level of prostaglandin F2α, for which drug analogs are known to be efficient in reducing IOP [180]. An FIV vector coding for the prostaglandin F synthase (PGF) was built under the control of a CMV promoter and injected (1 × 10^8^ TU) in a non-human primate (NHP) anterior chamber to reach the trabecular meshwork. Five NHPs were treated in one eye, with the contralateral eye receiving a GFP vector. The IOP was followed up for 550 days for certain animals. Directly after the injection, a weak reduction in IOP of about 2 mmg Hg was observed, and the pressure returned to normal levels after 5 months. Three animals received a second injection, but this treatment did not significantly affect IOP. One animal received a third dose on day 250 without inducing IOP change. Only one animal receiving one dose showed a decreased IOP of about 2 mm Hg at 5 months p.i. GFP monitoring also showed variation during the experiment. The trabecular meshwork appeared to correctly transduce 50 days p.i., but the expression decreased with time, even when a second injection was performed. The authors hypothesized that an innate resistance mechanism may occur, such as the presence of TRIM5α, which may sequester the viral capsid, preventing so the reverse transcription. This may explain why a second dose was not efficient but does not explain why the expression decreased after the first one. As suggested above, promoter shut-off or inefficient targeting of stem/progenitor cells at the origin of the trabecular meshwork may be responsible for this gene expression extinction.

Prostaglandin synthesis depends on cyclooxygenase-2 (COX2) activity, so the possibility of boosting PGF2 expression was explored using FIV-derived vectors coding for COX2 in combination with PGF or prostaglandin F receptor (FPR, [176]). To improve the transgene expression and prevent destabilization of the mRNA due to repetitive AUUA sequences, the UTR region was deleted. In addition, codon optimization was computed and led to a 10-fold increase in transgene expression with vectors containing optimized cDNA (RNA in reality) sequences compared to native cDNA. Prostaglandin yield in 293T cells transduced by the optimized FIV-COX2 was tremendously increased (from almost 0 to 35 ng/mL). Cat eye anterior chambers were transduced with 1 × 10^7^ TU in 200 µL after removal of a similar volume of aqueous humor. LV was well tolerated in general, but 3 animals out of 15 presented mild inflammation in the anterior chamber and conjunctiva that were well controlled by anti-inflammatory drugs (ketoprofen administered for 5 days). The most efficient vector combination to decrease IOP was obtained with those coding for COX2 and FPR, which induced a constant decrease of 5.4 mm Hg for 5 months (p = 0.03). Pressure reduction (about 35%) is in accordance with chronic treatment using drops of prostaglandin analogs (reduction of 25%, [181]). The authors also showed that no decrease in aqueous humor production occurred. These observations strongly demonstrate the pertinence of such an approach.

FIV vector transduction was improved in ex vivo *macaca mulatta* eye organ culture using a single injection of an MG132 proteasome inhibitor in the perfusion system one hour before the FIV injection [182]. MG132 final concentration in the anterior chamber was estimated to 20 µM. Ten to 20 µL of FIV-GFP (0.8 × 10^7^ TU or 2 × 10^7^ TU) were injected while eye perfusion was stopped for 1 h. The combined treatment increased both the level of GFP expression (GFP density) by almost five times and the number of genome copies by around two times. Histological analysis demonstrated that cells of the trabecular meshwork were transduced, suggesting that such a compound could help to maintain long-term expression by optimizing vector transduction in this tissue.

#### 2.4.2. Targeting the Retinal Ganglion Cells for Neuroprotection

Another approach consists of protecting RGCs, which form the optic nerve and die due to high IOP. To model RGC stress during glaucoma, axotomy or optic nerve crush was performed. Several neurotrophic factors were identified to have a survival effect on RGCs, such as BDNF, glial cell-derived neurotrophic factor (GDNF), ciliary neurotrophic factor (CNTF), and neurotrophin-4 (NT-4) [183,184,185,186], the most potent in vivo being CNTF [183]. To bypass the problematic of multiple injections to protect RGC, a SIN HIV-derived LV [187] was tested in a rat model of optic nerve axotomy [96]. This LV coding for CNTF had already shown its neuroprotective effect in a rat model of Huntington’s disease [188]. The evaluation of the LV tropism for RGC revealed that intravitreal injection of an LV-LacZ targeted around 25% of the RGCs, without inducing inflammations requiring supplemental medications. A single LV-CNTF intravitreal injection at the time of axotomy preserved 32% of the RGCs 21 days after the nerve transection when only 7% remained in the control group.

A similar approach was also performed to test the neuroprotective effect of PEDF. To model glaucoma-induced RGC stress in rats, acute high IOP was either induced by hypertension maintained for 60 min by balanced salt solution infusion through a cannula inserted in the anterior chamber or by intraocular injection of NMDA. An SIV was used to express hPEDF [79] (10 µL of 2.5 × 10^7^ TU/mL) and was subretinally injected 2 weeks prior to RGC injury [189]. A pressure of 110 mg Hg for one hour induced a reduction of about 60% of RGC number in the control SIV-empty vector-injected group, whereas a decrease of 44% was noticed in the SIV-hPEDF-injected group. A similar pattern of protection was obtained in the NMDA-eye-treated group.

These two studies show that LV-mediated gene transfer of neurotrophic factors in RGC or RPE cells results in an effective release of CNTF or PEDF to support RGC survival. Interestingly, one advantage of LV over many vectors is the rapid expression of the transgene, allowing the rescue of cells even when the vector is administrated along with the injury appearance in those models.

### 2.5. Gene Therapy Strategies to Prevent Corneal Fibrosis and Neovascularization

Several chronic diseases also affect the cornea and require repetitive treatments, which are usually well tolerated by the patient because of the easy access to this tissue. Nevertheless, some diseases need constant local treatment to optimize the modulation of the targeted molecular pathway. Gene therapy development for cornea has already been very well presented and discussed [190]. Here, we focus essentially on the strategy design of LV use for corneal fibrosis and cornea transplantation to enhance cell survival.

Stromal cells provide a crucial environment to maintain the fate and the integrity of epithelial and endothelial cells while being at the interface of these two monolayers. For instance, inflammation may transform corneal epithelial cells to epidermal cell fate [191]. Dysfunction of stromal cells may lead to endothelial cell junction permeabilization with, as a consequence, edema formation [192]. Cornea also secretes soluble Flt1-receptor to prevent the vascularization of the cornea from the conjunctiva [193]. Despite that topical drug treatment often restore a physiological situation, certain chronic alterations necessitate a deeper intervention to prevent sight loss. While cornea graft is very successful in eyes devoid of neovascularization and inflammation, these events reduce cornea transplantation success to about 50% (CCTS, 1992). Graft survival thus necessitates a tight control of the cornea environment and decreases in correlation with endothelial cell death [194].

In the cornea field, gene transfer was first developed to enhance the survival of transplanted mouse cornea tissue. More precisely anti-apoptotic gene transfer strategy was used to protect endothelial cells during grafting. Although anti-apoptotic factors may favor general tumor formation, such a strategy was used because endothelial cells are normally arrested in the G1 phase in the tissue. Cell death was promoted in a corneal endothelial cell line either by exposition to etoposide (DNA damage induction) or cytokines usually secreted during inflammations, such as TNKα and INFγ. Among the different LVs tested, LV-IZsGreen-xL (coding for Bcl-Xl and GFP) was found to have a protective effect by reducing by about 50% the number of Annexin-V-positive cells (dying cells), whereas LV-p35, LV-Bcl2, and LV-Survivin did not show a significant protective effect. In vivo, an allograft survival rate of about 90% at 8 weeks post transplantation was obtained by pre-incubating the graft with polybrene (6 µg/mL) and LV-IZsGreen-xL (1 × 10^7^ IU/mL), whereas the control vector or untreated group showed 30% and 40% of survival, respectively. Interestingly, this level of graft survival was obtained while only 15% of endothelial cells were transduced. This observation suggests a positive by-standard effect due to so far unidentified released factors.

To translate such technology to human applications, human cornea explants were challenged with the previously cited vectors (3 × 10^5^ IU/mL, [195]) before apoptosis induction by two chemical compounds, actinomycin and etoposide. Such treatment increased by around four times the number of Annexin-V-positive cells. The vector titer allowed to transduce more than 90% of corneal endothelial cells. Both LV-Bcl-xL and LV-p35 prevented more than 85% of endothelial cells from becoming TUNEL-positive cells, an observation in accordance with the transduction efficiency. Despite the argument of the authors that no tumor formation has been reported from corneal endothelial cells, the vector safety of Bclx-xL may be questionable since the proliferation of corneal endothelial cells can still be observed in vitro (for review [196]). Nonetheless, an inducing promoter would be an added value to these types of constructions involving apoptotic regulator genes. While this work must be seen as a proof of concept that endothelial cells can be protected by such an approach, long-term studies are still required, and the identification of a pathway less linked to cell transformation would be more appropriate to transition to humans. One target could be, for example, the cell death-initiating factors such as cytokines [197].

To counteract inflammation mediated by TNFα-secreting monocytes, an LV was designed to express the IL-10 [198], the most anti-inflammatory cytokine [199]. An MOI of 50 (based on endothelial cell count) for 4 days of the LV-SV40-IL10 was found to be optimal for ex vivo sheep cornea transduction to produce a sufficient amount of IL-10. The inhibitory potential of the released IL-10 to prevent TNFα secretion from monocytes was then assessed in cell line culture. After such validation, corneas were incubated with LV-SV40-IL10 for 3 h and transplanted into a sheep. The mock-treated group rejected the graft after a median of 18 days. Transplanted corneas were already colonized by neovessels on day 6. Similar observations were obtained with grafts incubated with the control vector LV-SV40-eYFP, whereas the therapeutic vectors prolonged the graft tolerance until 25 days post-implementation with graft neovascularization starting at day 9. Histological analyses revealed mononuclear cell invasion of all grafts in the stroma and the endothelium associated with a massive loss of this tissue. Although the LV-SV40-IL10 cornea transduction may be considered a modest gain of graft survival, it supports the paradigm that LV can release factors to control the cornea environment, but the most efficient factors remain to be identified, and the level of expression has to be optimized for this condition. Indeed, the use of an adeno-viral vector to transduce the cornea allowed a tremendous production of IL-10, up to more than 103 times those obtained with the LV vector in similar conditions, and prolonged the graft survival by 35 days [200] instead of 7 days with the LV. The use of WPRE to stabilize the expression and a non-viral promoter origin may have helped to enhance the production of the desired molecule.

Another candidate to reduce inflammation and prolong graft acceptance in allogenic conditions was tested with an LV vector coding for programmed death-ligand 1 (PD-L1 [201]). Rat corneas were transduced ex vivo with LV-PD-L1 (20 µL of a DMEM medium containing between 3.4 × 10^7^ and 1 × 10^8^ TU/mL). LV-eGFP-treated and non-treated corneas had a similar survival rate of 12.3 days ± 1.9 and 13.8 days ± 1.7, respectively. Interestingly, the LV-PD-L1-treated group protected the cornea by increasing the survival of 83% of them to the endpoint of the experiment (30 days). Corneas of this group also presented less opacity in comparison to the control groups suggesting less cell infiltration and less edema formation, which was then attested by slit lamp examination, histology, and flow cytometry analysis of infiltrating cells. Indeed, NKT, NK, and activated-NK cells as well as activated CD4+ T-cell population (CD3+CD8+CD161+, CD3–CD8+CD161+, CD3–CD8+CD161++, and CD4+CD134+ respectively) were present in controlled allogenic grafted corneas with the NKT cells being the most present. In the LV-PD-L1 group, this population was reduced by around 44% and the most affected by the treatment, suggesting their major contribution to graft rejection.

Since endothelial cells were often the main target of cell transduction to preserve the graft, a surgical procedure was developed to enhance potential transgene expression: a pocket in the stroma is created using a femtosecond laser followed by vector injection inside the pocket. Such an approach was tested in pig cornea explants with an HIV-derived vector expressing GFP (LV-CMV-GFP) and allowed stable GFP expression from 5 to 21 days in culture in four layers of stromal cells on the endothelial side and eight layers on the epithelial side [97]. The borders of the pocket were fully reattached during this period. Depending on the targeted tissue, endothelium, or epithelium, the pocket can be precisely performed at the proximity of the desired site. This approach, combined with endothelial cell transduction, for instance, may greatly enhance the control of the cornea environment to prevent either neovascularization or inflammation, or both when the grafting procedure appears not optimal.

In general, corneal epithelium or endothelium transplantations present a suitable success rate when conditions are optimum. The different gene therapy approaches presented here may enlarge the applications of transplantation in hosts for whom the cornea is already severely affected by inflammation or neovascularization. Nonetheless, progress in the understanding of rejection mechanisms is needed to identify the most effective anti-inflammatory factors.

## 3. General Perspectives

The different works described above demonstrated that lentiviral vectors are serious alternatives for gene transfer in the ocular paradigm. The high capacity of genome packaging and the long-term expression they can conduct in ocular tissue open new opportunities in ophthalmology to treat neovascularization, AMD, glaucoma, or inherited retinal diseases. As a comparison, while AAV vectors were also shown to drive stable expression in many ocular cell types, their therapeutic cassette is limited to less than 5 kb. The main strategy proposed to circumvent this limitation is the use of two complementary AAV vectors to eventually deliver larger transgenes. However, the efficiency of such an approach, which requires delivering two vectors without doubling their doses, still needs to be improved [202].

Many works also examined the safety of lentiviral vectors, which appear to have limited genotoxicity in non-dividing cells such as those in the ocular tissues. Nevertheless, as for other viral vectors, pre-existing immunity, in the case of lentiviral vectors only in HIV1-positive patients, could impair the success of gene therapy. However, pre-conditioning these patients to decrease their immune system activation could be a possible solution to still use this vector for therapy. As AAVs are non-pathogenic, their use as vectors seems, at first sight, more appropriate to avoid an immune response. Nevertheless, the pre-existing immunity in a large percentage of the population is a barrier for many capsids that were shown to be neutralized even in the eye. Alternative capsids were engineered to bypass the immune system, and anti-inflammatory treatments are still largely used for human delivery [203,204]. So far, RPE or trabecular meshwork targeting by lentiviral vectors is recognized as being very efficient in transferring therapeutic cassettes. More controversies exist regarding the transducing efficiency of photoreceptors, whereas AAV vectors efficiently target not only photoreceptors but also inner retinal cells depending on their capsids. We discussed several developments to redirect lentiviral vectors’ affinity to other cell types through envelop modifications. The ophthalmology field still has many opportunities to treat other diseases with the use of these strategies to readdress the selectivity of lentiviral vectors.

Finally, apart from the conventional gene therapy strategies such as gene replacement and neurotrophic support, editing of the host genome for gene correction or gene activation/inhibition based on the CRISPR/Cas9 system is also being tested using lentiviral vector tools. Even if the safety issue of such innovative approaches still has to cross some crucial steps before clinical application, these novel technological developments will certainly shape the future of gene therapy in ophthalmology.

## Figures and Tables

**Figure 1 pharmaceutics-14-01605-f001:**
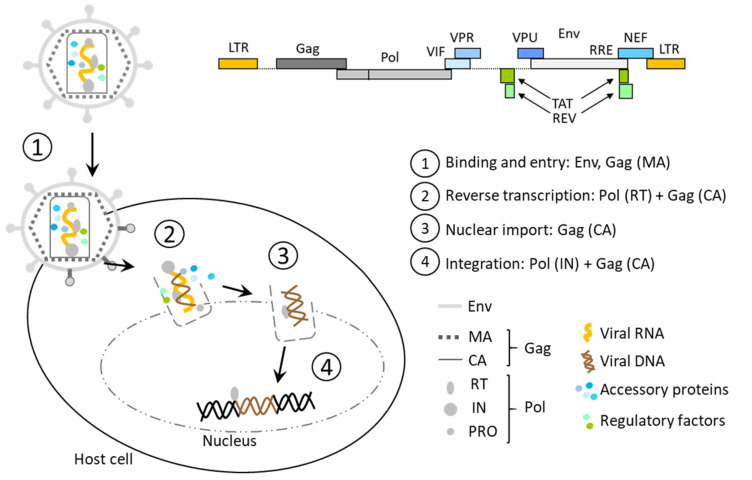
Entry and integration of HIV-1. Structural and enzymatic HIV-1 genes are presented in gray boxes, HIV-1 regulatory factors in green, HIV-1 accessory factors in blue, and LTR in yellow. Corresponding proteins are represented with the same color code in a schematic HIV-1 particle. The different processes, from binding to the cell membrane to the integration of the provirus DNA into the host genome, are depicted with viral protein required at each step. The timing of uncoating of the capsid (CA) and the role of CA is still unclear, as discussed in a recent review [11]. The Gag gene codes for matrix proteins (MA), capsid proteins (CA), and nucleocapsid protein (NC). The Pol gene encodes for the reverse transcriptase (RT), the integrase (IN), and the protease (PRO).

**Figure 2 pharmaceutics-14-01605-f002:**
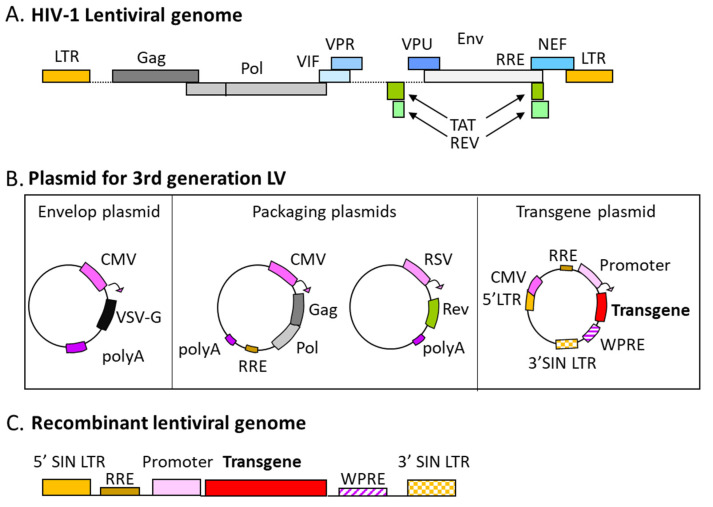
From lentivirus to 3rd generation lentiviral vector system. (**A**) Schematic representation of the natural HIV-1 genome. Structural and enzymatic HIV-1 genes are presented in gray boxes, HIV-1 regulatory factors in green, HIV-1 accessory factors in blue, and LTR in yellow. (**B**) Composition of the 4 plasmids co-transfected to generate LV; the envelope plasmid, the 2 packaging plasmids (expression of viral proteins), and the transgene plasmid (therapeutic cassette). Regulatory sequences used in the different plasmids generated for recombinant lentiviral production are in pink boxes, and the transgene in a red box. The curved arrow represents the start site of transcription. (**C**) Final genome of the recombinant lentiviral vector integrated into the host cell.

**Figure 3 pharmaceutics-14-01605-f003:**
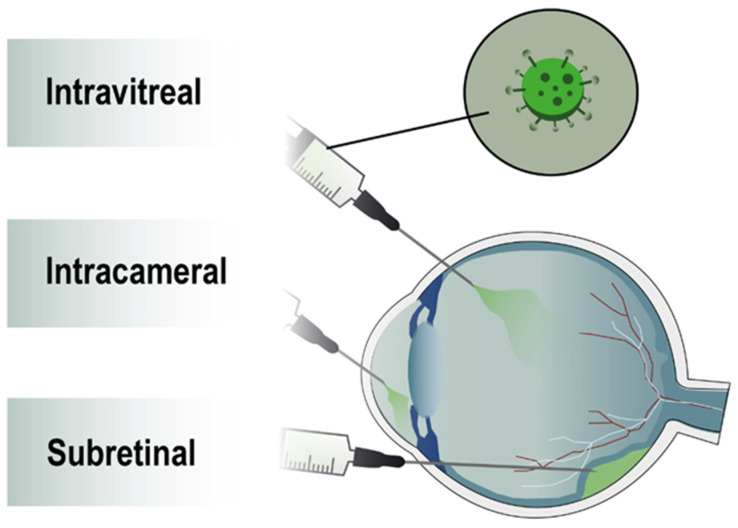
Ocular administration routes. Lentiviral vectors are delivered in the eye using three main routes: intravitreal injection in the vitreous humor to target the ganglion cell layer, intracameral injection in the aqueous humor to target the trabecular meshwork, and subretinal injection between the neuroretina and the RPE to reach RPE and photoreceptors.

**Figure 4 pharmaceutics-14-01605-f004:**
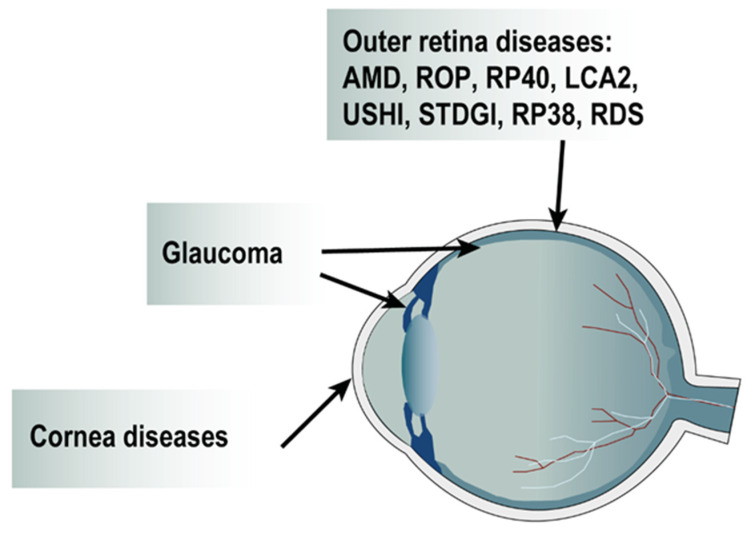
Diseases currently targeted by LV. LV treatments are evaluated for clinical application to treat several diseases of the anterior and posterior ocular compartments.

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
