# Peer review of "Lentiviral Vectors for Ocular Gene Therapy"

_pharmaceutics, 2022, doi:10.3390/pharmaceutics14081605_

Round 1
Reviewer 1 Report
This is a very comprehensive, well-written and timely review on lentiviral vectors and their application for ocular gene therapies.
The only critique that this reviewer has is that the authors could compare some results of the gene therapies obtained with lentiviral vectors to (the most commonly used) AAV vector. That would help the field to decide on which vector to focus their research.
There are just a few minor details that the authors could consider:
Lane 100: what do the authors mean with polyproline cis active sequence?
Lane 130: Gag encoding: p6 is missing, and NC could be spelt out (like Matrix and capsid)
Lane 186: “derive” – maybe not the best word? Perhaps “use” or “utilise” would be a better choise
Lane 203: “To determine, predict (…)” is an awkward sentence. Consider revising it?
Lane 268: “targets” – maybe could consider using “mediates” instead?
Lane 312: “Replacement of the promoter sequence” – for clarity, consider writing “replacement of the HIV-1 promoter sequence”
Reviewer 2 Report
Dear Authors,
Good review compiled. Comments are as follows:
1. Line 15 & 18 and so on: Avoid word We
2. Line 25: Keywords, As per my view ocular should be there in place of eye
3. Line 150 and 151: can be found here, remove this text. Author can cite the text where ever appropriate in above text
4. Line 153: Remove (Figure 1 ). Modify with :as depicted or shown in figure 1.
5. Check the sentence, grammar: Line 883,
